# Highly pathogenic avian influenza A(H5N1) virus in a common bottlenose dolphin (*Tursiops truncatus*) in Florida
Allison Murawski[1], Thomas Fabrizio [2], Robert Ossiboff [1], Christina Kackos [2], Trushar Jeevan[2], Jeremy C. Jones [2], Ahmed Kandeil [2], David Walker[2], Jasmine C. M. Turner [2], Christopher Patton [2,3], Elena A. Govorkova [2], Helena Hauck[1], Suzanna Mickey [1], Brittany Barbeau [4], Y. Reddy Bommineni[5], Mia Torchetti[6], Kristina Lantz [6], Lisa Kercher [2], Andrew B. Allison[1], Peter Vogel [7], Michael Walsh[1] & Richard J. Webby [2] ✉

Since late 2021, highly pathogenic avian influenza (HPAI) viruses of A/goose/Guangdong/1/1996 (H5N1) lineage have caused widespread mortality in wild birds and poultry in the United States. Concomitant with the spread of HPAI viruses in birds are increasing numbers of mammalian infections, including wild and captive mesocarnivores and carnivores with central nervous system involvement. Here we report HPAI, A(H5N1) of clade 2.3.4.4b, in a common bottlenose dolphin (*Tursiops truncatus*) from Florida, United States. Pathological findings include neuronal necrosis and inflammation of the brain and meninges, and quantitative real time RT-PCR reveal the brain carried the highest viral load. Virus isolated from the brain contains a S246N neuraminidase substitution which leads to reduced inhibition by neuraminidase inhibitor oseltamivir. The increased prevalence of A(H5N1) viruses in atypical avian hosts and its cross-species transmission into mammalian species highlights the public health importance of continued disease surveillance and biosecurity protocols.

There has been an explosive expansion in the geographic distribution and prevalence of highly pathogenic avian influenza (HPAI) A(H5N1) of hemagglutinin (HA) clade 2.3.4.4b viruses in animal populations in the Americas since 2021[1–3]. These viruses have raised concern due to their significant adverse economic impacts on the poultry industry and widespread mortality in wild bird species[1]. Notably, the virus has also been detected in a number of raptors such as eagles, owls, and vultures, suggesting predation and scavenging of infected birds are efficient means of HPAI virus transmission[1,2,4].

Recently, clade 2.3.4.4b A(H5N1) viruses have been responsible for mortalities in sea lions (*Otaria flavescents*) in Peru and Chile, along with harbor (*Phoca vitulina*) and gray (*Halichoerus grypus*) seals in New England and Canada[5–7]. Historically, multiple unusual mortality events in seal populations have been associated with avian influenza. Harbor seals (*Phoca vitulina*) appear to be the most susceptible marine mammal species for developing fatal respiratory disease, although the reason for this is currently unknown[8,9]. Many of these previous seal influenza-related mortality events that occurred in New England seal populations were caused by different influenza subtypes including A(H7N7) in 1979, A(H4N5) in 1983, A(H4N6) in 1991, A(H3N3) in 1992, and A(H3N8) in 2011[10–14]. Other significant outbreaks in seals have occurred in European waters (including A(H10N7) in 2014)[8,15–17].

In contrast to pinniped species, avian influenza viruses have rarely been detected in cetaceans and have not been associated with any unusual mortality events. Avian influenza viruses have been isolated from cetaceans including baleen whales (family *Balaenopteridae*)[13,18] such as the common minke whale (*Balaenoptera acutorostrata*)[19], as well as pilot whales (*Globicephala melaena*)[13,20,21], Dall's porpoise (*Phocoenoides dalli*)[19], and belugas (*Delphinapterus leucas*)[13,22].

The recent spread of clade 2.3.4.4b A(H5N1) viruses has been accompanied by novel infections of cetacean species around the world

[1]Department of Comparative, Diagnostic, and Population Medicine, College of Veterinary Medicine, University of Florida, Gainesville, FL 32608, USA. [2]Department of Infectious Diseases, St. Jude Children's Research Hospital, Memphis, TN 38105, USA. [3]Department of Microbiology, Immunology, and Biochemistry, University of Tennessee Health Science Center, Memphis, TN 38105, USA. [4]Department of Large Animal Clinical Sciences, College of Veterinary Medicine, University of Florida, Gainesville, FL 32608, USA. [5]Bronson Animal Disease Diagnostic Laboratory, 2700 N John Young Parkway, Kissimmee, FL 34745-8006, USA. [6]National Veterinary Services Laboratories, Animal and Plant Health Inspection Service (APHIS), U.S. Department of Agriculture (USDA), Ames, IA 50011, USA. [7]Comparative Pathology Core, St. Jude Children's Research Hospital, Memphis, TN 38105, USA. ✉e-mail: richard.webby@stjude.org

including three common dolphins (*Delphinus delphis*) in Peru, Wales, and England, two harbor porpoises (*Phocoena phocoena*) in Sweden and England, and an Atlantic white-sided dolphin (*Lagenorhynchus acutus*) in Canada[6,23]. In the harbor porpoise from Sweden, influenza virus was predominately found within the brain causing meningoencephalitis, consistent with this current report and other recent detections in terrestrial mammals[23]. The harbor porpoise displayed neurological signs including circling, inability to right itself, and subsequent drowning[23].

Similarly, spillover of clade 2.3.4.4b A(H5N1) in terrestrial mammals has also shown viral infection predominately in the brain, along with associated neurological signs. In 2006, a stone marten (*Martes foina*) naturally infected with a HPAI A(H5N1) virus developed neurological symptoms due to encephalitis, likely secondary to consumption of an infected wild waterfowl[24]. Other wild carnivore species have been diagnosed with HPAI A(H5N1), including a study of wild terrestrial mammals in the United States showing that 53 of 57 HPAI A(H5N1) virus-infected animals displayed neurological abnormalities[25].

Here we report an infection of HPAI A(H5N1) clade 2.3.4.4b in a common bottlenose dolphin (*Tursiops truncatus*) in Florida, United States, with the highest viral load in the brain causing meningoencephalitis. Adding to the existing literature on A(H5N1) viruses in cetacean species, the availability of tissues and multiple samples from this animal allowed us to combine a pathologic and virologic evaluation of the infection. Research was conducted to confirm the presence of A(H5N1) virus in dolphin tissues, biologically characterize the virus, and to determine the phylogenetic relationship between viruses from the dolphin and local wild bird populations. This study highlights the importance of combining histological lesions with virus isolation in order to better understand HPAI A(H5N1).

## Results

### Stranding response

On 29 March 2022, the Florida Fish and Wildlife Conservation Commission's Marine Mammal Pathobiology Laboratory (FWC MMPL) notified the University of Florida Marine Animal Rescue Program (UF MAR) of a dolphin in distress in a canal north of 178 4th Ave. West Horseshoe Beach in Dixie County, FL 32648 (29.44222°N, -83.29004°W). The complainant stated that the live dolphin was located between a seawall and dock piling, measuring approximately 20 centimeters wide, in the canal behind their home. The complainant attempted to free the dolphin while providing supportive care, however, the dolphin passed shortly after discovery and prior to the arrival of UF MAR. The dolphin was packed in ice and transported to the University of Florida College of Veterinary Medicine (UF CVM) on 29 March 2022 to be stored in a cooler for necropsy the following day.

### Gross and microscopic postmortem examination

At the UF CVM on 30 March 2022, the lungs were inflated and a computed tomography (CT) scan was performed, followed by a postmortem examination. The 191 cm juvenile, male common bottlenose dolphin (*Tursiops truncatus*; UFTt2203) was in thin body condition. Gross findings included an empty gastrointestinal tract, numerous superficial and deep linear epidermal lacerations, severe erosion of the caudal pharyngeal mucosa at the base of the larynx, mild pulmonary nematodiasis, lymphadenomegaly, and soft tissue proliferation at the tip of the penis.

On microscopic examination, the most prominent finding was moderate to marked mononuclear (dominated by lymphocytes and histiocytes) inflammation of the brain (Fig. 1a, b) and leptomeninges (Fig. 1c) with neuronal necrosis and neuropil malacia (Fig. 1b). The changes were the most predominant in the diencephalon, but also present within the mesencephalon and the brainstem. Meningeal and parenchymal vessels were frequently cuffed by mononuclear cells and there were multifocal areas of acute hemorrhage (Fig. 1a). Multifocal foci of mild gliosis were also present in the mesencephalon. These microscopic findings are all more suggestive of a longer, subacute, infection of the brain. The only observed

pulmonary changes included multifocal, mild, chronic granulomas and emphysematous change presumably secondary to postmortem insufflation (Fig. 1d). No viral inclusions or syncytial cells were noted in any examined tissues. No microorganisms were visualized in brain sections with histochemical stains (Gomori-methenamine silver or periodic acid-Schiff). Other microscopic findings unrelated to the central nervous system lesions included a squamous penile papilloma, verminous laryngitis with intralesional nematodes, gastric trematodes, and generalized reactive lymphoid hyperplasia.

### Ancillary diagnostics

Samples of the affected brain were screened for the presence of apicomplexan parasites (such as *Toxoplasma gondii* and *Sarcocystis* spp.) and cetacean morbillivirus by PCR and RT-PCR, respectively, at the UF CVM Zoological Medicine Diagnostic Laboratory; both tests were negative. Lung and brain tissue samples were subsequently sent to the Bronson Animal Disease Diagnostic Laboratory (a member of the National Animal Health Laboratory Network, accredited for testing for Select Agents) to screen for HPAI viruses and eastern equine encephalitis virus. HPAI virus was detected in both the lung and brain, with the brain having a higher viral load (Table 1). No other viruses were detected. Tissue samples were then sent to the National Veterinary Services Laboratories (NVSL) for official confirmatory testing (Table 1). Real-time RT-PCR (rRT-PCR) assays were then used to test for influenza A virus of Eurasian A/goose/Guangdong/1/1996 (H5N1) lineage. The sequence of HA cleavage site, along with sequencing of NA gene, were used to characterize the virus subtype and pathotype. The virus was confirmed as HPAI A(H5N1) virus of HA clade 2.3.4.4b.

### Tissue distribution of H5N1 viral RNA

Dolphin tissues collected prior to confirmation of the presence of HPAI A(H5N1) virus were transferred to an approved Biosafety Level 3 enhanced (BSL3-enhanced) laboratory at St. Jude Children's Research Hospital, Memphis, TN. Real-time RT-PCR (rRT-PCR) was used to detect the presence of H5 virus in selected tissues using influenza A virus matrix (M)- and H5 gene-specific primers. Brain and spinal cord homogenates were positive for influenza virus with $C_T$ values of <30 when probed for both M and H5 RNA. Tissue from the brain had the highest concentration of viral RNA with a M gene $C_T$ value of 15.7 and H5 $C_T$ value of 23.6. Influenza virus RNA was less prevalent in the lung with a $C_T$ value of 32.5 when H5 primers were used. Interestingly, gastric samples had a M gene $C_T$ value of 28.9 and H5 was detected with a $C_T$ value of 34.2. Trace M gene RNA was also detected in liver and kidney samples ($C_T > 34$), suggesting low-level systemic dissemination of the virus (Table 2).

### Immunohistochemistry

To further evaluate the tissue sites for presence of HPAI A(H5N1) viral RNA in the dolphin, immunohistochemistry (IHC) was performed on lung and brain tissues. IHC revealed viral antigen present in perivascular neurons and neuropil throughout the central nervous system (CNS; Fig. 2a). In situ hybridization (ISH) confirmed these results. As expected, viral RNA was localized in the same areas of the CNS as was viral antigen (Fig. 2b). At higher magnification, IHC and ISH detected viral antigen and RNA in the neuronal soma and axonal processes (Fig. 2c, d). However, when lung sections were stained, weak IHC labeling was present in some alveolar surfaces and within the pulmonary interstitium; these sections of lung were negative for viral RNA by ISH suggesting that virus infection was not present in the lung (Fig. 2e, f). Additionally, sporadic virus positive cells were detected in circulation, as well as the heart and lung, suggesting low grade viremia. These sporadic viral positive cells could not be definitively identified due to their indistinct morphology, and the significance at this time is unknown. Multiple levels of the gastrointestinal tract (forestomach, pyloric stomach, glandular stomach, proximal small intestine, middle small intestine, and colon) were also evaluated for the presence of influenza virus nucleoprotein by IHC. No definitive positive cells were detected in the

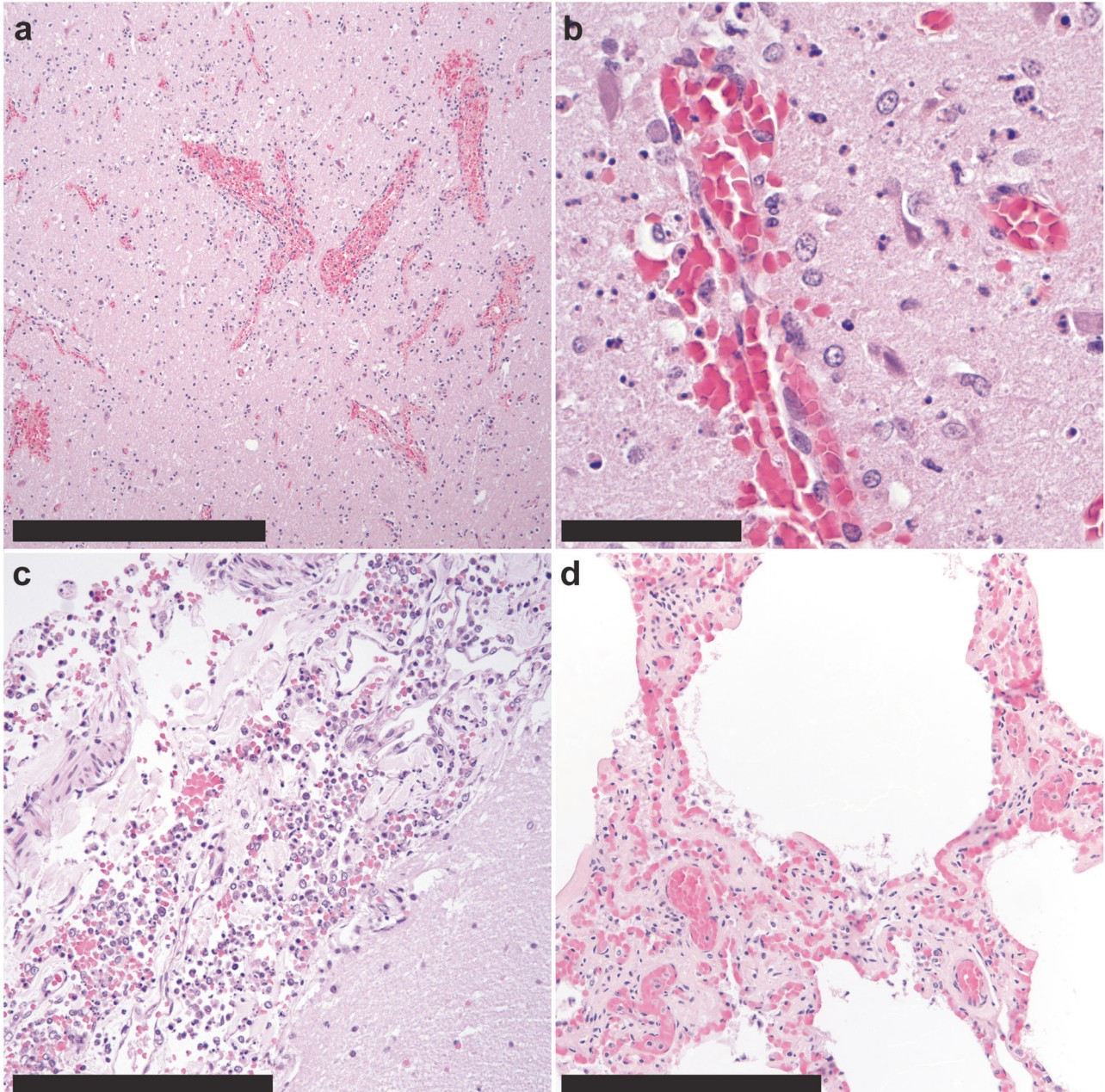

**Fig. 1 | Microscopic features of influenza A(H5N1) virus infection in a common bottlenose dolphin (*Tursiops truncatus*). a** In the brain, the cerebral neuroparenchyma is hypercellular with areas of congestion, hemorrhage, and loose, mononuclear perivascular cuffs; bar = 500 μm. **b** Within the affected neuropil, there is frequent necrosis of neurons and glial cells, as characterized by cytoplasmic eosinophilia and nuclear fragmentation; bar = 50 μm. **c** The leptomeninges are expanded by mixed mononuclear cells and hemorrhage; bar = 250 μm. **d** The pulmonary parenchyma was largely unremarkable, changes consistent with mild postmortem autolysis; bar = 250 μm. Hematoxylin and eosin, all panels.

intestinal mucosal. Gastrointestinal myenteric plexi were present and evaluated at all levels; they were all negative for virus antigen. Overall, the widespread presence of A(H5N1) virus antigen in the CNS and the absence of detection in the respiratory tissues, as well as systemic diffuse virus positive cells, by IHC and ISH are consistent with the rRT-PCR results.

**Virus isolation**

Virus isolation was performed on lung and brain tissue, and two isolates (A/bottlenose dolphin/Florida/UFTt2203/2022) were obtained after one passage of homogenized brain tissue in embryonated chicken eggs as well as in MDCK cells. No isolates were grown from the lung tissue, even after multiple blind passages in either embryonated chicken eggs or MDCK cells.

**Genome sequencing and phylogenetic analysis**

Sequence analysis of the full genome of A/bottlenose dolphin/Florida/UFTt2203/2022 (H5N1) virus was obtained directly from the brain homogenate as well as from virus isolated in embryonated chicken eggs and MDCK cells inoculated with homogenates from the dolphin brain. The genomic sequences from all sources were identical at the amino acid level. Sequences have been deposited in GenBank under accession numbers OP698125-OP698132. As expected, all eight gene segments of A/bottlenose dolphin/Florida/UFTt2203/2022 (H5N1) virus clustered with viruses isolated from birds and wild mammals in North America (data for HA shown in Fig. 3). The virus was identified as a genotype B1.2 reassortant with polymerase basic 2 (PB2; clustered with am1.1), polymerase basic 1 (PB1; clustered with am1.1), polymerase acidic (PA; clustered with am1), and

**Table 1 | Viral load in the lungs and brain of dolphin**

| Dolphin's tissue | Influenza A virus (M primers) | H5 HA clade 2.3.4.4b (HA primers) | A(H5N1) virus (H5 primers) |
|---|---|---|---|
| Lung | 34.95 / 35.1 | 36.84 / 34.8 | 35.07 / 32 |
| Brain | 16.58 / 22.2 | 17.28 / 23.0 | 16.28 / 19.4 |

rRT-PCR $C_T$ values of lung and brain dolphin tissues reported by the Bronson Animal Disease Diagnostic Laboratory (BADDL) and National Veterinary Services Laboratories (NVSL). RNA extracted from the brain and lung tissues of dolphins were screened using influenza A virus M- and H5-specific primers and probes to test for the presence or absence of HPAI A(H5N1) virus by rRT-PCR. BADDL results are listed first and NVSL results are listed second. Numbers represent $C_T$ values of selected tissues determined by rRT-PCR.

**Table 2 | Viral load in the selected tissues of dolphin**

| Dolphin's tissue | Influenza A virus (M primers) | A(H5N1) virus (H5 primers) |
|---|---|---|
| Brain | 15.7 | 23.6 |
| Spinal Cord | 22.0 | 26.1 |
| Lung | 27.8 | 32.5 |
| Liver | 34.3 | – |
| Kidney | 36.2 | – |
| Gastric | 28.9 | 34.2 |

rRT-PCR $C_T$ values of influenza A matrix and H5 RNA isolated from dolphin tissues. RNA extracted from brain, spinal cord, lung, liver, kidney, and gastric samples was screened using influenza A virus M- and H5-specific primers and probes to test for the presence or absence of HPAI A(H5N1) virus by rRT-PCR. The $C_T$ value of the highest dilution of the controls was determined to be the limit of detection and served as the threshold for determining whether samples were PCR positive or negative. Numbers represent the $C_T$ values of selected tissues determined by rRT-PCR.

nucleoprotein (NP; clustered with am1.2) genes derived from North American lineages with the remaining genes of Eurasian lineage origin (clustered with ea1) (Fig. S1). All gene segments had at least 99% identity to those derived from other viruses detected across North America and determined to be genotype B1.2 which circulated along the Atlantic and Mississippi flyways during the spring of 2022 with steady detections even into the Central flyway but did not reach the Pacific flyway. Notably, we also sequenced the genomes of HPAI A(H5N1) viruses isolated from deceased birds including from lesser scaup (*Aythya affinis*), common loon (*Gavia immer*) and a laughing gull (*Leucophaeus atricilla*) from the same geographic area (Dixie County, Florida) and approximate same time frame (March 2022). This coincided with outbreaks infecting local wild bird species. The genotypes of these viruses differed from the genotype of the dolphin virus suggesting that they were not direct precursors. Similarly, viruses isolated from an A(H5N1) outbreak in seals in the Northeast United States were unreassorted genotype A2 and not related to the dolphin isolate ruling out these animals as a source of infection.

Sequence data was examined for presence of markers associated with mammalian adaptation. Only an HA-T192I (H3 numbering) amino acid substitution in the 190-helix situated adjacent to the receptor binding site was identified, which has been shown to increase receptor α2,6-sialic acid binding[26]. The impact of this substitution on sialic acid specificity is unknown in this A(H5N1) genetic backbone. Additionally, NA-S246N (N2 numbering), a substitution associated with reduced susceptibility to neuraminidase inhibitor (NAI) oseltamivir, was present in A/bottlenose dolphin/Florida/UFTt2203/2022 (H5N1) virus. No markers associated with resistance to adamantanes or endonuclease inhibitors were identified[27,28]. Similarly, PB2 substitutions that are often associated with mammalian infection of avian influenza viruses were absent[29].

**Antiviral susceptibility**

Due to the NA-S246N substitution, A/bottlenose dolphin/Florida/UFTt2203/2022 (H5N1) virus was evaluated for susceptibility to NAIs. Oseltamivir susceptibility of A/bottlenose dolphin/Florida/UFTt2203/2022 (H5N1) virus was 18-fold lower than for N1-subtype matched oseltamivir susceptible reference virus A/Denmark/524/2009 (H1N1)pdm09, with $IC_{50}$ values of 7.51 nM vs. 0.41 nM respectively (Fig. 4a, c). A/bottlenose dolphin/Florida/UFTt2203/2022 and reference virus A/Denmark/524/2009 (H1N1)pdm09 had similar $IC_{50}$ values for zanamivir (0.52 nM vs. 0.31 nM, respectively) and peramivir (0.72 nM vs. 0.10 nM, respectively) (Fig. 4a, c). For reference, A/Denmark/528/2009 (H1N1) pdm09 containing NA-H274Y (N2 numbering) was also evaluated and displayed highly reduced inhibition by oseltamivir and peramivir with $IC_{50}$ values of 155.0 nM and 23.5 nM, respectively (Fig. 4b). Therefore, A/bottlenose dolphin/Florida/UFTt2203/2022 (H5N1) containing NA-S246N displayed reduced susceptibility to oseltamivir at the low end of the inhibition scale as classified by World Health Organization (WHO) guidance for NAI susceptibility reporting. The virus retained full susceptibility to peramivir and zanamivir.

**Receptor binding properties**

Due to the HA-T192I substitution (H3 numbering) that was unique in A/bottlenose dolphin/Florida/UFTt2203/2022 (H5N1) virus and supposed to be in the 190-helix situated adjacent to the receptor binding pocket, receptor binding assay was examined. The current H5N1 isolate preferred to bind sialic acid receptors with a 3′SLN-linked sialic acid (avian influenza viruses' receptors), rather than 6′SLN-linked sialic acid receptors (human influenza viruses' receptors) (Fig. 5).

**Antigenic characterization**

To examine the antigenic properties of A/bottlenose dolphin/Florida/UFTt2203/2022 (H5N1) virus, hemagglutination inhibition assays were conducted with reference viruses and antisera. A/bottlenose dolphin/Florida/UFTt2203/2022 (H5N1) virus was antigenically indistinguishable from other 2021/2022 clade 2.3.4.4b A(H5N1) viruses collected in North America. They reacted well with post-infection ferret antiserum generated against A/Astrakhan/3212/2020 (H5N1), the WHO's reference virus for this clade.

**Discussion**

This report represents the first known detection and isolation of influenza A virus from a common bottlenose dolphin (*Tursiops truncatus*), found deceased on 29 March 2022. Prior to necropsy and microscopic evaluation, influenza was not considered as a differential diagnosis for this stranded bottlenose dolphin. However, the discovery of meningoencephalitis on microscopic evaluation led us to consider neuroinvasive etiologies previously diagnosed in other Florida wildlife. HPAI virus was considered as a potential etiology because HPAI (H5N1) virus-infected wild birds commonly have encephalitis on postmortem examinations[30], and in conjunction, there had been numerous reports of wild bird die-offs secondary to HPAI virus in Florida (notably many waterbird species such as ducks, gulls, and terns). In addition, in 2022, recent mortalities of harbor seals and gray seals in Maine, United States were diagnosed with HPAI A(H5N1), with a small subset of this population displaying neurological signs[5]. Since the initial introduction of the clade 2.3.4.4b A(H5N1) into North America in 2021, these viruses have spread and affected numerous avian and mammalian species across the United States[1,2]. This raised the possibility there could have been spillover into another marine species (common bottlenose dolphin) with neurological involvement. This suspicion was subsequently verified by the detection and isolation of A(H5N1) virus in the brain tissues of the dolphin.

Previous in vitro research has shown minimal to no attachment of avian A(H4N5) and A(H7N7) avian influenza viruses to the trachea and bronchi of bottlenose dolphins, suggesting that they had low susceptibility to influenza viral infections[14,31]. This research is similar to the results from our

study, as there was limited viral detection of HPAI in the lungs. Our findings showed the highest viral load within brain tissues with limited evidence of systemic dissemination of the virus. Overall, more research is needed to determine the true susceptibility and route of exposure of influenza viruses

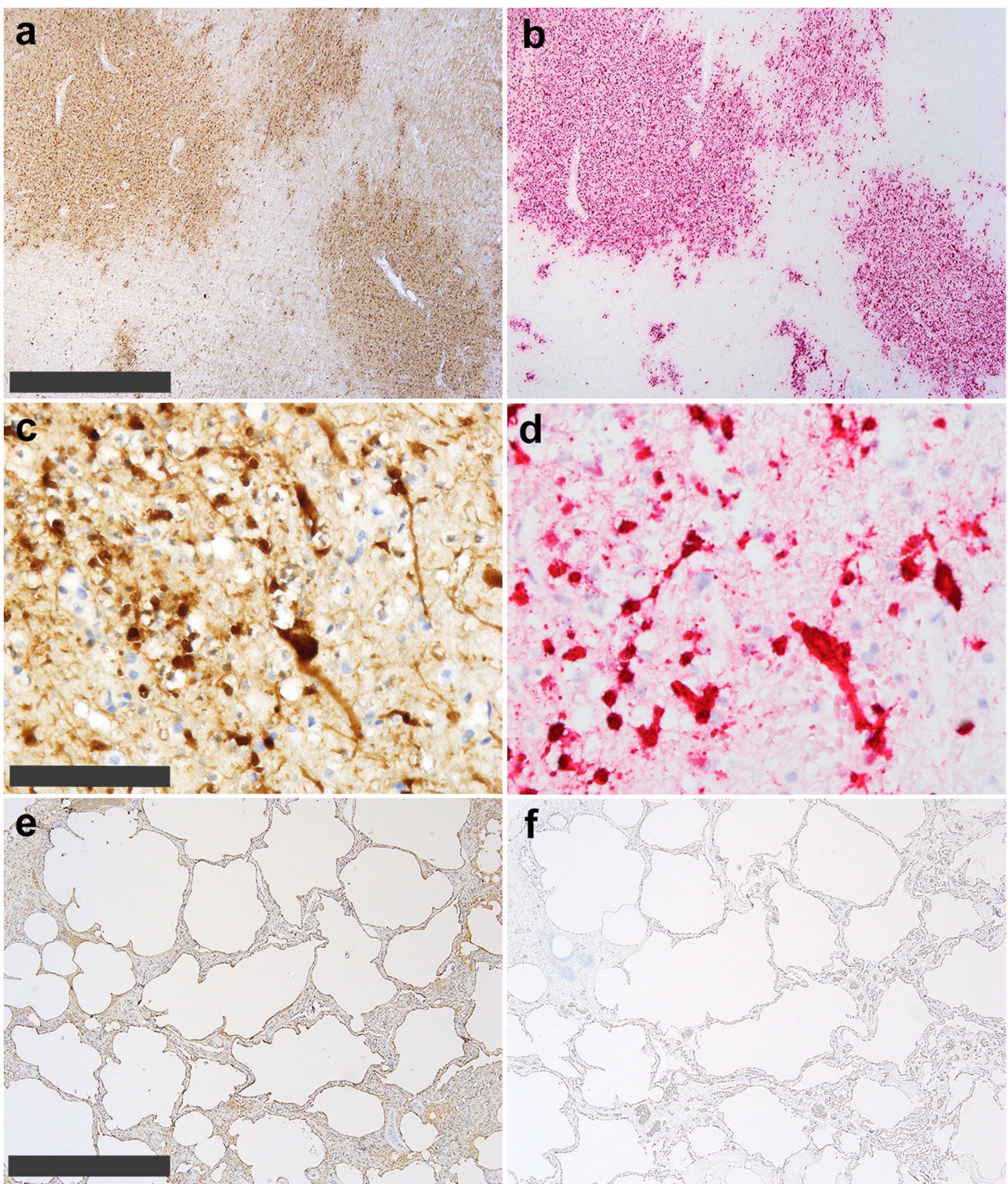

**Fig. 2 | Detection of influenza A(H5N1) virus antigen and virus RNA in dolphin tissues by immunohistochemistry (IHC) and in situ hybridization (ISH) in the brain and lung. a** Numerous antigen-positive foci in perivascular neurons and neuropil throughout the CNS by IHC. **b** Localization of influenza virus RNA in the same areas of CNS by ISH in sequential sections. **c** Higher magnification showing virus antigen in the neuronal soma and axonal processes by IHC. **d** Localization of virus RNA in the same cellular locations by ISH. (**e**) Indeterminate weak IHC labeling of lung tissue along some alveolar surfaces and within pulmonary interstitium. (**f**) ISH on sequential sections of lung were all negative for virus RNA. Brown staining = IHC detection of viral antigen (**a, c, e**); Red staining = ISH detection of viral RNA (**b, d, f**); Scale bars: Brain A and B = 2 mm; Brain **c** and d = 100 µm; Lung **e** and **f** = 1 mm.

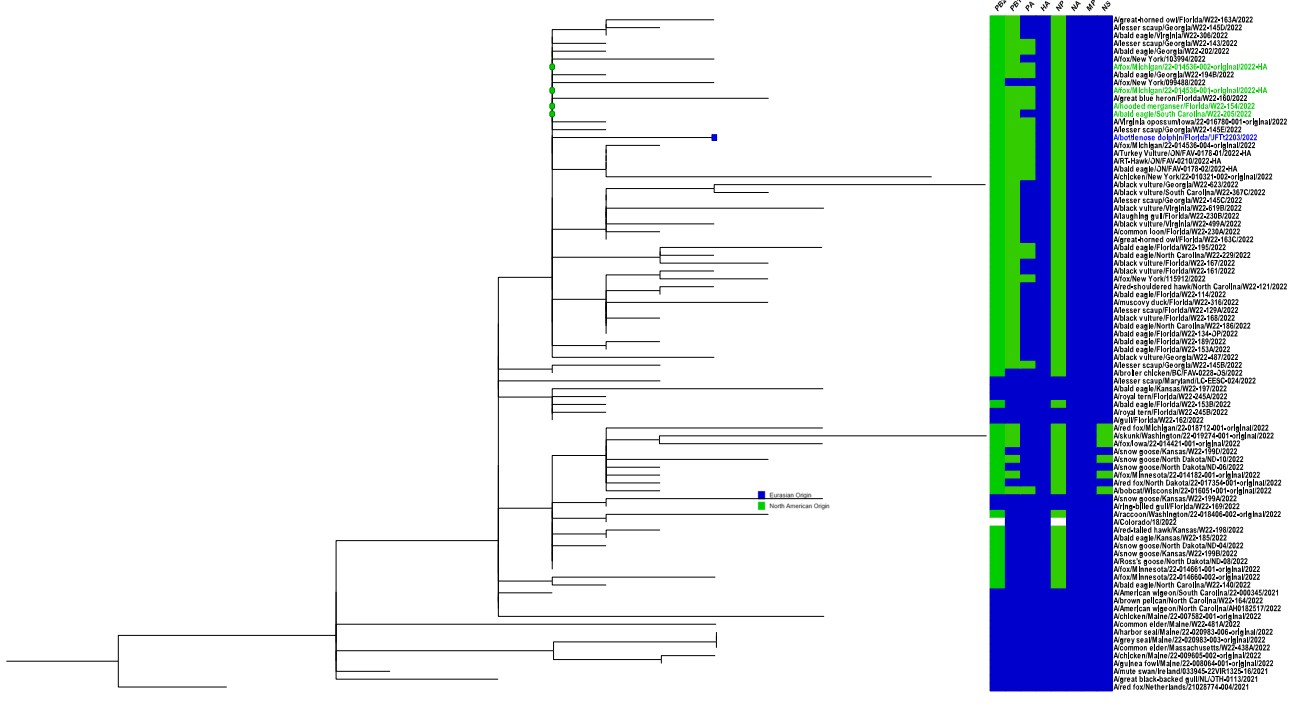

**Fig. 3 | Phylogenetic alignment of influenza A/bottlenose dolphin/Florida/ UFTt2203/2022 (H5N1) HA gene sequences with Eurasian and North American lineage viruses.** Phylogenetic tree representing HA segments primarily from viruses collected from North American birds inferred using neighbor joining methods displayed with all 8 gene segment origins. The HA from the virus identified in the bottlenose dolphin is represented by the blue square while the green circles indicate the most similar sequences identified which differ by three nucleotides. Gene segments originating from North American viruses are represented in green, Eurasian origin gene segments are colored blue, and white indicates no data available.

in cetaceans. Since cetaceans were previously thought to have low susceptibility to influenza viral infections, this study emphasizes the importance of routinely testing for HPAI viruses in all mammals and obtaining proper samples, including brain, during postmortem examinations.

The localization of the virus to the CNS in the dolphin is consistent with other HPAI A(H5N1) viral infections in mammals. HPAI virus-infected wild red foxes (*Vulpes vulpes*) in the Netherlands had CNS involvement with no detection in the respiratory tract or other organs[32]. Likewise, a study examining 40 HPAI A(H5N1) viruses in Canada found clinical presentation to be primarily neurological in red foxes, striped skunks (*Mephitis mephitis*), and mink (*Neovison vison*)[33]. In the case of an infected stone marten (*Martes foina)*, it was believed that the animal had already cleared its respiratory infection and the only notable remaining lesion was its encephalitis[24]. This raises the possibility that the dolphin may have had pulmonary involvement that was cleared before the neurological manifestations and subsequent death had occurred – however, this is unlikely as no histological evidence of pulmonary disease was found to support this idea. In addition, experimental intranasal inoculation of HPAI A(H5N1) in ferrets (*Mustela putorius furo*) results in encephalitis through infection of the olfactory mucosa spreading to the CNS via the olfactory and trigeminal nerves[34]. In contrast, most toothed whales (odontocetes), with the exception of baleen whales (mysticetes), lack olfactory anatomy, eliminating the possibility of olfactory tract neuroinvasion by influenza virus in dolphins[35]. Instead, a primary hematogenous route of entry into the CNS is suspected, with secondary neuron-to-neuron invasion, suggested by the apparently random distribution and extensive spread of virus infection in the brain. Systemic viral spread may have occurred via direct infection from the intestinal lumen. A study of H5N1 experimentally-infected cats demonstrated ganglioneuritis in the submucosal and myenteric plexi of the small intestine, indicating the potential for direct viral spread via the intestines, especially when respiratory tract disease is absent[36]. On necropsy, the dolphin had severe ulcerative inflammation of the pharynx with extensive loss of epithelium extending from the caudal tongue past the larynx. Histologically these areas demonstrated moderate to severe mucosal autolysis and no evidence of virus was found. All gastrointestinal sections were also negative for influenza virus nucleoprotein by IHC, so the port of entry for the virus remains unclear in this case. More research needs to be done to determine how the virus spreads to the CNS in this dolphin.

The United States Department of Agriculture (USDA) confirmed at least 10 seabird die-offs related to HPAI A(H5N1) virus during March 2022 in Dixie County, FL between Horseshoe Beach and the mouth of the Suwannee River, where the dolphin was found. The potential exposure route of this dolphin to avian sources of HPAI is speculative, though these birds were thought to be a potential source of infection for this dolphin. However, comparative sequencing from available viruses demonstrated that these bird viruses were not direct precursors. Similarly, the virus isolated from the dolphin was not directly descended from viruses detected in seals in the Northeast United States, ruling out a dominant lineage circulating in marine mammals. The typical route of transmission to carnivorous mammals (via ingestion of infected birds) is thought to be unlikely in marine mammals because birds are not a typical food source for seals or dolphins[1,2,4,5]. However, there have been instances of dolphins and killer whales in facilities killing birds, such as sea gulls. These instances would increase their exposure to potentially infectious agents. In the wild, killer whales are known to kill and eat birds such as penguins and other sea birds. Williams et al.[37] divided killer whale interactions with birds into 3 categories: play, predation, and interaction[37] with some encounters similar for smaller cetaceans. Humpback whales (*Megaptera novaeangliae*) are known to take birds into the oral cavity when seabirds are feeding on the same prey items and genetic evidence of accidental ingestion of Murrelets (*Synthliboramphus antiquus* and *Brachyramphus marmoratus*) has been described in humpback feces[38]. In addition, some dolphins may strand feed (pushing fish onto muddy banks where they can be ingested). Birds are often in close proximity taking advantage of this technique, therefore fecal and urate exposure would increase. The close

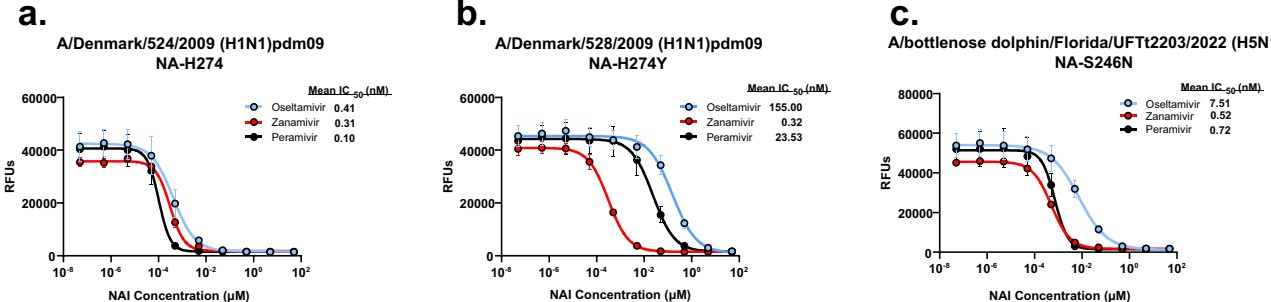

**Fig. 4 | NAI susceptibility of A/bottlenose dolphin/Florida/UFTt2203/2022 (H5N1) virus. a** NAI susceptibility of A/Denmark/524/2009 (H1N1)pdm09 NA-H274 (N2 numbering) wild-type reference virus. **b** NAI susceptibility of A/Denmark/528/2009 (H1N1)pdm09 NA-H274Y oseltamivir/peramivir-resistant reference virus. **c** NAI susceptibility of A/bottlenose dolphin/Florida/UFTt2203/2022 (Dolphin/FL) virus. Classifications follow WHO guidance for NAI susceptibility evaluation.

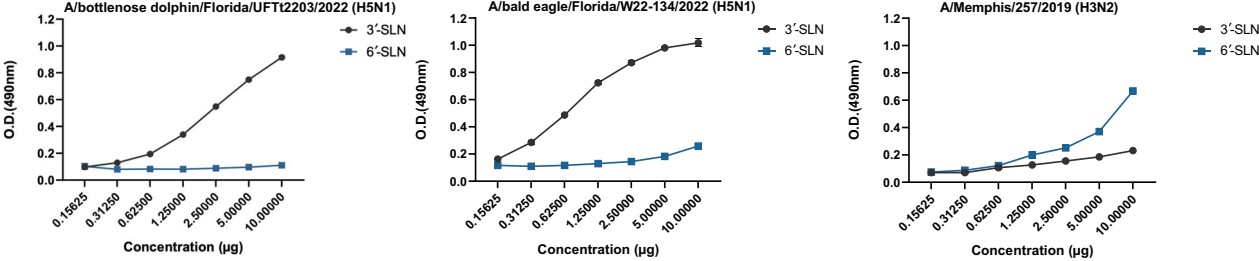

**Fig. 5 | Receptor-affinity assay of the A/bottlenose dolphin/Florida/UFTt2203/2022 (H5N1) virus.** Direct binding of the virus to sialylglycopolymers (3'-SLN or 6'-SLN) at different concentrations (X axis) was measured at optical density (O.D.) of 490 nm (Y axis). A/bald eagle/Florida/W22-134/2022 (H5N1) and A/Memphis/257/2019 (H3N2) were included as control viruses.

proximity of dolphins and sea birds during communal feeding on schools of fish increases the potential exposure from fecal and urates exposure orally, and though highly unlikely, through the external nares above the water surface during inhalation. Birds infected with HPAI may shed infectious virus in their saliva, mucous, and feces[1,2,4,5]. Therefore, environmental transmission, or through mouthing/biting or inquisitive investigation of the bird in water, may be a likely route of transmission for marine mammals through oral, nasal, or conjunctival routes[5,39]. However, strand feeding is not a known feeding method used in the area in which this dolphin was found due to the presence of oyster bars; although it is possible the dolphin may have accidentally come in contact with an infected bird while feeding on jumping fish, or while feeding on fish near the surface of the water.

Similar to this study, historically, cetaceans infected with low-pathogenic influenza A have displayed secondary neurological signs. For example, a pilot whale recovered off the coast of Maine in 1984 was diagnosed with A(H13N2) and A(H13N9) after demonstrating difficulty swimming and diving[20]. Similarly, just prior to the dolphin stranding, the UF MAR received an unverified report of a dolphin swimming erratically in the vicinity of the stranding, suggesting that this dolphin could have displayed neurological signs prior to death. Currently, due to the widespread prevalence of HPAI virus within wildlife populations in North America, any cetacean (as well as other marine or terrestrial mammal) displaying neurological signs, whether typical or atypical hosts of influenza, should be considered for testing for HPAI virus.

According to the United States Centers for Disease Control and Prevention (US CDC), the current health risk of A(H5N1) viruses to humans is relatively low due to the inability for sustained human-to-human transmission[2]. In this case, replication of the avian A(H5N1) virus in the dolphin did not lead to the acquisition of obvious mammalian adaptive markers. We did, however, note a HA-T192I substitution. This substitution has not been seen in any other 2.3.4.4b A(H5N1) virus sequenced from the Americas (using sequences available in GISAID as assessed 1 March 2023). The impact of this substitution on binding to cellular receptors is unclear,

although it has been shown in one study to enhance binding of virus with HA-T192I to the mammalian virus preferred α2,6 sialic acids[26] but had no impact on binding in another[40]. More research is needed to understand the impact of the HA-T192I substitution in this dolphin isolate. The other marker of potential relevance to human health was an NA-S246N. This substitution led to a reduced susceptibility to oseltamivir in vitro. How this reduced susceptibility translates to clinical impact is unknown, although it might be anticipated that the drug would still have utility. Additionally, A/bottlenose dolphin/Florida/UFTt2203/2022 (H5N1) virus is susceptible to other NAIs (zanamivir and peramivir) and PA protein inhibitor (baloxavir marboxil).

Human health risk aside, the consequences of A(H5N1) viruses adapting for enhanced replication in and transmission between dolphins and other cetacea could be catastrophic for these populations. The molecular markers associated with influenza virus adaptation to these hosts is unknown, and, therefore, it is not possible to determine if there is any evidence of it occurring. Enhanced follow up of naturally occurring cetacea A(H5N1) infections with viral sequencing is prudent, with the possibility of identifying common mutations suggestive of host range markers.

The detection of HPAI A(H5N1) virus in a bottlenose dolphin emphasizes the importance of following proper biosafety protocols when working with stranded marine mammals. As shown in this case, high viral loads were detected in brain tissues, underscoring the value of obtaining brain tissues during routine postmortem marine mammal examinations[39]. With no end in sight for the current A(H5N1) outbreak, testing for avian influenza viruses should be a part of future routine health monitoring in marine mammal stranding investigations, especially if the animal displays any neurological signs.

## Methods
### Necropsy and histopathology
A complete gross necropsy of dolphin was performed on 30 March 2022. Sections of the following tissues were retained frozen (−80 °C): lung, liver,

brain, kidney, spinal cord, cerebrospinal fluid, muscle, skin, and a penile papilloma. A complete set of tissues were collected in 10% neutral buffered formalin: right cortex at caudate nucleus, right diencephalon at thalamus, right mesencephalon and hippocampus, right brainstem and cerebellum, spinal cord, liver, spleen, kidney, right and left adrenals, cranial omentum with debris, thyroid, right and left cranial lung, right and left middle lung, right caudal lung, right distal lung, trachea, tracheal lymph nodes, right and left tonsils, mandibular lymph nodes, cranial esophagus, laryngeal mucosa, laryngeal accessory lymph node, thymus, right and left atria, right and left ventricles, pulmonary artery, pulmonary marginal lymph node, inter-ventricular septum, aorta, diaphragm, blubber and skin, urinary bladder, lumbar vertebral lymph node, glandular soft tissue, pancreas, tongue, pharynx, forestomach, pyloric stomach, glandular stomach, proximal intestine, pulmonary lymph hilar lymph node, middle intestine, mesenteric lymph node, distal intestine, colonic lymph node, prescapular lymph node, melon, testis, penis, right and left pterygoid sinus mucosa, and skeletal muscle and nerve. Representative sections of all tissues were collected, submitted for routine processing, and 5 μm sections were cut and stained with hematoxylin and eosin (HE). Additional histochemical stains (Gomori-methenamine silver and periodic acid-Schiff) were utilized on select tissue sections to look for the presence of microorganisms.

## Virus isolation

A/bottlenose dolphin/Florida/UFTt2203/2022 A(H5N1) virus was isolated from brain tissue homogenate by inoculation of 10-day old embryonated hens' eggs incubated at 35 °C for 40 h, and from inoculation of Madin-Darby canine kidney (MDCK) cells (ATCC-CCL-34) incubated at 37 °C and 5% CO$_2$ for 60 h. All were subsequently harvested and stored at −80 °C pending sequence confirmation. All isolation work was performed in approved Biosafety Level 3 enhanced (BSL3-enhanced) laboratories at St. Jude Children's Research Hospital, Memphis, TN. The virus used in this study is a USDA-classified select agent and is subject to the guidelines and compliance with Title 9 (CFR Parts 121 [Possession, Use, and Transfer of Select Agent Toxins] and 122 [Importation and Transportation of Controlled Organisms and Vectors]).

## Nucleic acid extraction and real time RT-PCR

Viral RNA was extracted from the following tissues using Qiagen RNeasy® Mini kit (Qiagen, Hilden, Germany) per the manufacturer's protocol: brain, spinal cord, kidney, lungs, liver, penis papilloma, muscle, and gastric contents. Freshly extracted RNA was screened using H5-specific primers and probes to test for the presence or absence of HPAI A(H5N1) via real time RT-PCR (rRT-PCR) according to a protocol established by the US CDC. The definition of rRT-PCR positivity was established based on 10-fold serial dilutions of copy-number control H5 RNA down to approximately one copy number. The C$_T$ value of the highest dilution of the controls was determined to be the limit of detection and served as the threshold for determining whether samples were PCR positive or negative. The sequences of primers and probes were obtained from the US CDC (CDC Ref# I-007-05).

## Hemagglutination inhibition assays

Once A(H5N1) sequence was confirmed, the antigenic properties of this virus were compared to other recently isolated North American A(H5N1) viruses and A(H5) HA clade 2.3.4.4 antigens and sera as recommended by the WHO Global Influenza Surveillance and Response System (GISRS). Reference post-infection ferret antiserum was used after treatment with receptor-destroying enzyme II (Denka Seiken Co.) at 37 °C overnight, heat inactivated at 56 °C for 45 min. Assays were conducted with 0.5% chicken red blood cells as described in the WHO Global Influenza Surveillance Network Manual for the laboratory diagnosis and virological surveillance of influenza[41].

## Immunohistochemistry (IHC) and in situ hybridization (ISH)

Viral antigen and RNA were detected via immunohistochemistry (IHC) and in situ hybridization (ISH), respectively. IHC was performed using the Ventana Discovery Ultra Autostainer (Roche Ventana, Tucson, Arizona,

USA) following the manufacturer's instructions. FFPE sections (5 μm) were initially heated to 72 °C for 4 min and placed in EZ prep solution (#950-102, Roche Ventana) for deparaffinization. Antigen retrieval was performed at 95 °C in Cell Conditioning Solution 1 (CC1, #950-124, Roche Ventana) for 56 min. A polyclonal primary goat antibody (US Biological, Swampscott, MA) against influenza A/USSR/1977 (H1N1) nucleoprotein at 1:1000 and a secondary biotinylated donkey anti-goat antibody (catalog number sc-2042; Santa Cruz Biotechnology, Santa Cruz, CA) at 1:200 on tissue sections. The DISCOVERY ChromoMap DAB detection kit (#760-159, Roche Ventana) was used as the detection system. Tissue counterstaining was performed with Hematoxylin II solution (#790-2208, Roche Ventana). 6-to-8 week-old female BALB/c H5N1 infected mouse lung was used as the positive control tissue and normal goat serum as the negative control primary antibody. The normal goat serum control was used to rule out any non-specific binding of goat antibodies to formalin-fixed dolphin tissues and no evidence of non-specific binding with the normal goat serum was noted. The results of IHC labeling were then confirmed with a virus-specific ISH assay. To detect the influenza viral RNA in FFPE tissues, ISH was performed using the RNA scope 2.5 HD RED kit (Advanced Cell Diagnostics) according to the manufacturer's instructions. Briefly, 40-ZZ ISH probes (catalog 1048229-C1, V-influenzaA-H5N8-m2m1-C1) targeting influenza RNA were designed and synthesized by Advanced Cell Diagnostics. Tissue sections were deparaffinized with xylene, underwent a series of ethanol washes and peroxidase blocking, and were then treated using kit-provided antigen retrieval buffer and proteinase. ISH signal was developed using kit-provided pre-amplifier and amplifier conjugated to alkaline phosphatase and incubated with a fast red substrate solution for 10 min at room temperature. Sections were then counterstained with hematoxylin. The universal negative control probes developed by ACDbio, which target the DapB gene (accession # EF191515) from *Bacillus subtilis* strain SMY, a soil bacterium were used as controls in ISH.

## Genome sequencing and phylogenetic analysis

Using extracted viral RNA as template, cDNA was synthesized using the Uni-12 primer[42] with the Superscript IV First-Strand Synthesis System (Invitrogen). The influenza A virus gene segments were amplified using modified universal primers in a multi-segment PCR as previously described[43]. PCR products were then purified using Agentcourt AMPure XP beads according to the manufacturer's protocol (BeckmanCoulter). Sequence libraries were prepared using the Nextera DNA Prep Kit (Illumina), according to the manufacturer's protocol, and sequenced using the iSeq Reagent Kit v2 (300-cycles) on the iSeq System (Illumina). Sequencing reads were then quality trimmed and assembled using CLC Genomics Workbench *(version 22.0.2)* (Qiagen).

For the phylogenetic analysis, sequences other than those generated for the study were retrieved from the National Center for Biotechnology Information Influenza Virus Sequence Database and the EpiFlu database of the Global Initiative on Sharing All Influenza Data (GISAID). The HA sequences were aligned and trimmed to length based on the full HA coding region using CLC Genomics Workbench version 22.0.2 (Qiagen). The phylogenetic tree of the HA segment was constructed using the neighbor-joining method and Jukes-Cantor substitution model. Genotype data for each gene segment was then determined by BLAST analysis and overlayed with phylogeny using CLC Genomics Workbench version 22.0.2. The GenoFLU tool (https://github.com/USDA-VS/GenoFLU) was used for virus genotyping[44]. MEGA X was used for the construction of eight phylogenetic trees by applying the Neighbor-Joining method and Kimura 2-parameter model with 1000 bootstrap replicates for aligned and trimmed nucleotide sequences[45].

## Phenotypic susceptibility to NAIs

Virus susceptibility to the NAIs oseltamivir carboxylate (oseltamivir), zanamivir, and peramivir (MedChem Express) was determined by fluoro-metric assay with the substrate 2′-(4-methylumberlliferyl)-α-D-N-acet-ylneuraminic acid (MUNANA) (Sigma) as described[46], with minor

modifications. Equivalent amounts of NA activity for each virus (equal to the fluorescence signal from 10 μM 4-methylumbelliferone) were incubated with $\log_{10}$ serial dilutions of NAI (50 μM to 5 pm) at 37 °C for 30 min, followed by the addition of 100 μM MUNANA for an additional 30 min, 37 °C in assay buffer [32.5 mM MES, 4 mM $CaCl_2$, pH 6.5]. The fluorescence signal of the NA-cleaved substrate was measured at Ex/Em 360 nM/460 nM on a Synergy H1 microplate reader (Biotek). The 50% inhibitory concentrations ($IC_{50}s$) were estimated from dose-response curves using the sigmoidal, four-parameter logistic non-linear regression equation in GraphPad Prism (v9). The results are representative of the median values of four independent dose-response curves. Influenza A/Denmark/524/2009 (H1N1)pdm09 (NA H274, NAI susceptible) and A/Denmark/528/2009 (H1N1)pm09 (NA H274Y, NAI highly reduced susceptibility) were included as reference viruses. Fold-over the N1 subtype matched A/Denmark/524/2009 (H1N1)pm09 virus was used to interpret NAI-susceptibility phenotypes as recommended by the WHO Global Influenza Programme reporting guidelines (https://www.who.int/teams/global-influenza-programme/laboratory-network/quality-assurance/antiviral-susceptibility-influenza/neuraminidase-inhibitor); (<10-fold, normal inhibition; 10–100 fold, reduced inhibition; >100-fold, highly reduced inhibition).

### Phenotypic susceptibility to adamantanes and endonuclease inhibitors

Susceptibility to M2 protein inhibitors (amantadine, rimantadine) and PA protein endonuclease inhibitor baloxavir marboxil (BXM) was determined from full-length gene sequences. The M2 splice product from the full-length M gene was analyzed for the presence of M2 substitutions (L26F, V27A, A30T, S31N, or G34E) characteristic of adamantane resistance[27]. The PA endonuclease domain (residues 1-200) was analyzed for the presence of PA substitutions (E23G/K, I38T/F/M/S/L/N, and/or E119D) characteristic of BXM resistance[28].

### Receptor binding assay

Fetuin coated 96-well plates were washed three times using a washing buffer (0.23X PBS with 0.01% Tween 80). To prevent nonspecific binding, the coated plates were blocked using 1X PBS containing 1% BSA then the volume of 100 ul of A/bottlenose dolphin/Florida/UFTt2203/2022 (H5N1), A/bald eagle/Florida/W22-134/2022 (H5N1), or A/Memphis/257/2019 (H3N2) viruses having 64 HA units were added to each well and incubated overnight at 4 °C. Each viral sample was tested in duplicate. Biotinylated sialylglycopolymers, 3'-SialLacNAc-PAA-biotin (3'-SLN) or 6'-SialLacNAc-PAA-biotin (6'-SLN) (Glycotech), were serially diluted in the reaction buffer (1X PBS with 0.02% Tween 80, 0.02% BSA, and 5 μM oseltamivir carboxylate) then added to washed plates having viruses and incubated for 2 h at 4 °C. Plates were washed and horseradish peroxidase-conjugated streptavidin (1:2000) was added to the plates and incubated for 1 h. Following another round of washing, o-phenylenediamine dihydrochloride (Sigma-Aldrich) was added to the plates and kept at room temperature for developing color then the reaction was terminated by adding 1 N $H_2SO_4$. The absorbance of the plates was measured at an optical density (O.D.) of 490 nm.

### Statistics and data reproducibility

Neuraminidase susceptibility data were derived from four separate dose-response curves consisting of 10 serial $\log_{10}$ dilutions of each drug. Receptor binding data was derived from two independent experiments consisting of duplicate repeat measures of each the 7 ligand dilutions. These source data files are provided in the supplementary data file.

### Reporting summary

Further information on research design is available in the Nature Portfolio Reporting Summary linked to this article.

### Data availability

Data generated in this study are provided in the main manuscript, the Supplementary Fig. and table, and in source data files contained within the supplementary data. Sequences have been deposited in GenBank under accession numbers OP698125-OP698132, NCBI Biosample accession SAMN40758915. GISAID sequence accession numbers are provided in Table S1.

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

## Acknowledgements

The authors thank the University of Florida's Marine Animal Rescue Program (UFMAR), including Hannah Walsh (HW) and Hada Herring, University of Florida Aquatic Animal Health Program's veterinary students and volunteers, St. Jude Children's Research Hospital Department of Infectious Disease and World Health Organization Collaborating Center for Studies on the Ecology of Influenza in Animals and Birds, Bronson Animal Disease Diagnostic Laboratory, National Veterinary Services Laboratories (Ames, Iowa), Florida Fish and Wildlife Conservation Commission, Florida Department of Agriculture and Consumer Services, Southeastern Cooperative Wildlife Disease Study at the University of Georgia, and veterinary radiologists from the University of Florida, including Aitor Gallastegui. We gratefully acknowledge the National Marine Fisheries Service (NMFS) and Deborah Fauquier DVM for guidance. Stranding response by UFMAR is authorized under the Southeast Region of the Marine Mammal Health and Stranding Response Program under a NMFS stranding agreement. We also appreciatively acknowledge all data contributors, i.e., the authors and their originating laboratories responsible for obtaining the specimens, and their submitting laboratories for generating the genetic sequence and metadata and sharing via the GISAID Initiative, on which this research is based. This project has been funded in part with Federal funds from the National Institute of Allergy and Infectious Diseases, National Institutes of Health, Department of Health and Human Services, under contract 75N93021C00016, U.S. National Science Foundation (1911955) award, the University of Florida Aquatic Animal Health Program, and by ALSAC. The findings and conclusions in this publication are those of authors and should not be construed to represent any official USDA or U.S Government determination or policy.

## Author contributions

All authors actively participated in revision and discussion of this manuscript prior to submission. S.M. and H.W. participated in stranding response; M.W., S.M., B.B., H.H., and R.O. participated in necropsy diagnostics; A.A. recommended testing for HPAI; Y.R.B. and M.T. detected and confirmed presence of HPAI; R.W., L.K., M.W., A.M. conceptualized research project and methodologies; R.W., T.F., T.J., J.J., A.K., D.W., J.T., C.P., E.G., P.V., L.K., and R.W., acquired additional data; A.M., and C.K., wrote initial draft with contributions from all authors; A.M., R.W., R.O., A.A., and M.W., reviewed and edited subsequent drafts, along with applying input from all other authors. Funding was acquired by R.W., and M.W.

## Competing interests

The authors declare no competing interests.
