## [Peer Review File · Communications Biology]

Reviewers' comments:

Reviewer #1 (Remarks to the Author):

The manuscript of Murawski et al describe a case report of a stranded bottlenose dolphin that most likely died due to an encephalitis caused by Highly pathogenic H5N1 infection. The authors claim this is the first report of a bottlenose dolphin. The case occurred March 2022, and therefore was earlier than two others described in Peru, that occurred after November 2022, and are still in Biorxiv, although to me it is more relevant that this is the only record with a quite detailed histopathologic description of the lesions associated with the infection, as well as the viral sequence and other characteristics. The case is of interest to others in the community and the wider field. The current H5N1 situation with widespread replication in so many different bird species, with such high mortality is of huge concern for wildlife conservation, as well as potential of adaptation to mammals and subsequent spread in mammals, including humans. The behaviour of H5N1 and its spilling over to mammals has been shown to be unpredictable. When we learn more about what species are susceptible and what species are not, it could help to determine what host factors are important for susceptibility and predict the behaviour of influenza viruses in the future. Therefore I feel that the paper will influence thinking in the field.

I find the work convincing. I appreciate the use of a combination of virology, immunohistochemistry and in situ hybridization to evaluate the virus behavior in this novel host species.

To me it is important to better understand which route the bottlenose dolphin might have acquired brain infection. I think the discussion on this could be extended (see more detailed feedback on this in the attached file).

I have some minor comments on the manuscript, which are outlined in the attached file.

Reviewer #2 (Remarks to the Author):

In the manuscript entitled "Highly pathogenic avian influenza A(H5N1) virus in a common bottlenose dolphin 2 (*Tursiops truncatus*) in Florida", Dr Murawski and colleagues provide a detailed report about a common bottlenose dolphin in Florida that was infected with HPAI A(H5N1) clade 2.3.4.4b. Some basic information was already provided in the press release (A first: Avian influenza detected in American dolphin » College of Veterinary Medicine » University of Florida (ufl.edu)), but this reports provides an interesting and thorough description of the pathological findings, PCR results, phylogenetic analysis, NAI susceptibility and receptor affinity. Furthermore, experiments are well performed, although the phylogenetic analysis is somewhat basic.

A few minor comments: in the legend of figure 3, it is unclear why some virus names are indicated in green. Furthermore, it might be of interest to provide as supplemental figures phylogenetic analysis of the different gene segments of A/bottlenose dolphin/Florida/UFTt2203/2022 and closely related viruses. In addition, considering the very high viral loads in the brain, were any attempts performed to obtain the complete genome directly from the clinical sample?

**RESPONSES TO REVIEWER'S COMMENTS ON MANUSCRIPT
COMMSBIO-23-2140-T**

**Highly pathogenic avian influenza A(H5N1) virus in a common bottlenose dolphin
(*Tursiops truncatus*) in Florida**

Allison Murawski, Thomas Fabrizio, Robert Ossiboff, Christina Kackos, Trushar Jeevan, Jeremy C. Jones, Ahmed Kandeil, David Walker, Jasmine C. M. Turner, Christopher Patton, Elena A. Govorkova, Helena Hauck, Suzanna Mickey, Brittany Barbeau, Reddy Bommineni, Mia Torchetti, Lisa Kercher, Andrew B. Allison, Peter Vogel, Richard J. Webby, Michael Walsh

Reviewer #1 (Remarks to the Author):

The manuscript of Murawski et al. describe a case report of a stranded bottlenose dolphin that most likely died due to an encephalitis caused by Highly pathogenic H5N1 infection. The authors claim this is the first report of a bottlenose dolphin. The case occurred March 2022, and therefore was earlier than two others described in Peru, that occurred after November 2022, and are still in Biorxiv, although to me it is more relevant that this is the only record with a quite detailed histopathologic description of the lesions associated with the infection, as well as the viral sequence and other characteristics. The case is of interest to others in the community and the wider field. The current H5N1 situation with widespread replication in so many different bird species, with such high mortality is of huge concern for wildlife conservation, as well as potential of adaptation to mammals and subsequent spread in mammals, including humans. The behavior of H5N1 and it's spilling over to mammals has been shown to be unpredictable. When we learn more about what species are susceptible and what species are not, it could help to determine what host factors are important for susceptibility and predict the behavior of influenza viruses in the future. Therefore, I feel that the paper will influence thinking in the field.

I find the work convincing. I appreciate the use of a combination of virology, immunohistochemistry and in situ hybridization to evaluate the virus behavior in this novel host species.

COMMENT #1. To me it is important to better understand which route the bottlenose dolphin might have acquired brain infection. I think the discussion on this could be extended (see more detailed feedback on this in the attached file).

RESPONSE: The reviewer raises a very interesting point here; how did the virus get to the brain of this animal? We also consider this of particular importance and while we by no means have a definitive answer we have added the following information to the discussion.

“In contrast, most toothed whales (odontocetes), with the exception of baleen whales (mysticetes), lack olfactory anatomy, eliminating the possibility of olfactory tract neuroinvasion by influenza virus in dolphins⁵³. Instead, a primary hematogenous route of entry into the CNS is suspected, with secondary neuron to neuron invasion, suggested by the apparently random distribution and extensive spread of virus infection in the brain. Systemic viral spread may have occurred via direct infection from the intestinal lumen. A study of H5N1 experimentally-infected

cats demonstrated ganglioneuritis in the submucosal and myenteric plexi of the small intestine, indicating the potential for direct viral spread via the intestines, especially when respiratory tract disease is absent⁵⁴. On necropsy, the dolphin had severe ulcerative inflammation of the pharynx with extensive loss of epithelium extending from the caudal tongue past the larynx. Histologically these areas demonstrated moderate to severe mucosal autolysis and no evidence of virus was found. Multiple levels of the gastrointestinal tract (forestomach, pyloric stomach, glandular stomach, proximal small intestine, middle small intestine, and colon) were also evaluated for the presence of influenza virus nucleoprotein by IHC. No definitive positive cells were detected in the intestinal mucosal. Gastrointestinal myenteric plexi were present and evaluated at all levels; they were all negative for virus antigen. More research needs to be done to determine how the virus spread to the CNS in this dolphin.”

COMMENT #2. General introduction: I think the introduction contains a good overview of the current status of knowledge on spillover infections of avian influenza into marine mammals, but I think it is good to emphasize the value of combining the detection of the virus together with the detection of lesions and its association with virus. For the previous bottlenose H5N1 detections (ref23) lesions were not described.

RESPONSE: This is a worthy point to highlight and to the reviewers point we have now emphasized “*Adding to the existing literature on A(H5N1) viruses in cetacean species, the availability of tissues and multiple samples from this animal allowed us to combine a pathologic and virologic evaluation of the infection*” in the revised introduction.

COMMENT #3. Line 41: Regarding spread to the America’s of H5N1. Please consider if this article: Transatlantic spread of highly pathogenic avian influenza H5N1 by wild birds from Europe to North America in 2021 Scientific Reports (nature.com) is an additional good & necessary reference.

RESPONSE: We thank the reviewer for pointing out this omission. We have now added this reference.

COMMENT #4. Line 107: ‘other systemic microscopic findings’ It is unclear what is meant with systemic. E.g. Squamous penile papilloma is a localized microscopic finding. Consider omitting the word ‘systemic’. Gross and microscopic postmortem examination: I would favor and welcome one step of interpretation made extra here, and then a different way of writing these results. I would start with the macroscopic and microscopic findings that authors think are directly related to the H5N1 infection, then those indirectly related to the H5N1 infection, and lastly those that authors judged as being common ‘background’ lesions, not likely directly or indirectly related to the H5N1 infection. E.g. the papilloma and the parasitic lesions. In this way it is clearer for those readers unfamiliar with free-ranging cetacean pathology, how to interpret these results.

RESPONSE: We thank the reviewer for the comment. We have now added the following in response as well as omitting the word systemic.

“Other microscopic findings unrelated to the central nervous system lesions included a squamous penile papilloma, verminous laryngitis with intralesional nematodes, gastric trematodes, and generalized reactive lymphoid hyperplasia.”

COMMENT #5. Line 129 heading Tissue distribution → consider changing this heading to ‘tissue distribution of H5N1 viral RNA’

RESPONSE: We have made this change as suggested.

COMMENT #6. Line 140: please change virus → to viral RNA. In my experience viral RNA in tissues and samples is frequently detected while we cannot detect the virus with immunohistochemistry, in situ hybridization or culture, even though much effort was put into it. I realize that could be because of low nr of positive cells and only a small piece in the IHC slide; alternatively, it can be contamination or circulating small RNA fragments. As these findings are yet not well explained, I would be careful to interpret viral RNA fragment detection as viral presence, when this can not be confirmed via other methods.

RESPONSE: We agree with the reviewer here and have made the changes as suggested. This section now reads *“To further evaluate the tissue sites for presence of HPAI A(H5N1) viral RNA in the dolphin, immunohistochemistry (IHC) was performed on lung and brain tissues.”*

COMMENT #7. Line 151: sporadic cells in circulation → please describe the morphology of the antigen positive cells and then add the most likely ‘inflammatory’ cell type according to authors, based on the morphology. The authors suggest sloughed of endothelial cells might be the cell type, but no other endothelial cells (still part of blood vessel wall lining) tested positive – at least this is not described? If so, I think that is less likely that those cells are endothelial cells. Then, what type of inflammatory cell? Macrophage?

RESPONSE: Unfortunately, we have been unable to more definitively identify these cells. In response to the reviewers comments we have added the following *“Additionally, sporadic virus positive cells were detected in circulation, as well as the heart and lung, suggesting low grade viremia. These sporadic viral positive cells could not be definitively identified due to their indistinct morphology, and the significance at this time is unknown.”*

COMMENT #8. Line 239-241: This dolphin may have been infected with HPAI virus from interactions with deceased, sick, or dying birds struggling in the water, however the route of transmission is still currently unknown. This discussion is expanded in paragraph starting line 274. Consider to delete the sentence here for getting a better structure in the discussion. Rather than references of birds infected and excreting virus, it would additionally be helpful to know if bottlenose dolphins have been observed to have any of these play/mouth/eat interactions with birds. Any reports or literature on this? Is strand feeding observed in this species? Also related to risks for people swimming in water close to sick birds it is important to understand what kind of exposure could have led to this infection. Any information /references on how well this virus survives in seawater?

RESPONSE: In response to the reviewers suggestion we have deleted this sentence and added the following discussion on possible transmission routes.

*“The typical route of transmission to carnivorous mammals (via ingestion of infected birds) is thought to be unlikely in marine mammals because birds are not a typical food source for seals or dolphins¹⁻⁴. However, there have been instances of dolphins and killer whales in facilities killing birds, such as sea gulls. These instances would increase their exposure to potentially infectious agents. In the wild, killer whales are known to kill and eat birds such as penguins and other sea birds. Williams et al (1990) divided killer whale interactions with birds into 3 categories: play, predation, and interaction⁵⁵ with some encounters similar for smaller cetaceans. Humpback whales (*Megaptera novaeangliae*) are known to take birds into the oral cavity when seabirds are feeding on the same prey items and genetic evidence of accidental ingestion of Murrelets (*Synthliboramphus antiquus* and *Brachyramphus marmoratus*) has been described in humpback feces⁵⁶. In addition, dolphins in this area may also strand feed (pushing fish onto muddy banks where they can be ingested). Birds are often in close proximity taking advantage of this technique, therefore fecal and urate exposure would increase. The close proximity of dolphins and sea birds during communal feeding on schools of fish increases the potential exposure from fecal and urates exposure orally, and though highly unlikely, through the external nares above the water surface during inhalation. Birds infected with HPAIV may shed infectious virus in their saliva, mucous, and feces¹⁻⁴. Therefore, environmental transmission, or through mouthing/biting or inquisitive investigation of the bird in water, may be a likely route of transmission for marine mammals through oral, nasal, or conjunctival routes^{4,16}.”*

Unfortunately, we were not able to find any references pertaining to how well this virus survives in seawater.

COMMENT #9. Paragraph line 252: In my view an important paragraph ‘The localization of the virus to the CNS in the dolphin is consistent with other HPAI A(H5N1) viral infections in mammals.’ It is interesting to consider how the findings fit with the route to the CNS in cetacea as the pharynx and nose anatomy is so different from those of other mammals described infected with CNS involvement (mostly carnivores). Please see Influenza A virus (H5N1) infection in cats causes systemic disease with potential novel routes of virus spread within and between hosts - PubMed (nih.gov). Different patterns were observed depending on the route of inoculation.

RESPONSE: See response to the general comment #1 above. The listed reference has now been utilized and referenced in the manuscript.

COMMENT #10. Intestine was not tested with IHC, so maybe good to explain why (autolysis?) and how therefore neuronal plexi in the intestine could not be investigated, and the similarities of this case in the bottlenose to the cats inoculated via food, could not be further investigated. Or if possible, please test the intestines, to see if antigen can be traced in neuronal plexi in the intestines – or if these were necrotic, inflamed, or lost. Authors state in results that neurons surrounding blood vessels were positive in three different parts of the brain, suggesting virus might have come in via the circulation. Was distribution in the CNS seemingly random? Or localized to particular areas? Any likely inter neuronal route was followed by the virus? (were

neurologically connected areas involved – showing neuron to neuron transmission might have occurred? Or rather multiple sites of introduction in the brain where closely associated neurons became infected). As far as I am aware there is no olfactory mucosa in cetacea – I would like to see this back in the discussion so the make the reader realize that this is quite intriguing and a field that needs exploration. Also, authors note the inflammation observed was ‘mononuclear (dominated by lymphocytes and histiocytes) inflammation of the brain’. Although this fits with the harbor porpoise with H5N1 brain infection, in my experience the changes of HPAI infection in the brain is usually with neutrophils and necrosis rather than mononuclear (see descriptions of experimentally infected ferrets and cats). What does this mean according to the authors? Does this suggest the brain infection might have been there for a longer period of time prior to death? I would like to see these differences discussed.

RESPONSE: Despite the moderate to severe autolysis of mucosal epithelium evident in tissue sections, we evaluated multiple levels of the gastrointestinal tract (forestomach, pyloric stomach, glandular stomach, proximal small intestine, middle small intestine, and colon) for the presence of influenza virus nucleoprotein by IHC. No definitive positive cells were detected in the intestinal mucosal. Gastrointestinal myenteric plexi were present and evaluated at all levels; they were all negative for virus antigen. The perivascular inflammatory cell infiltrates in the brain and meninges consist predominantly of mononuclear in this dolphin. There are rare granulocytes scattered in parenchymal areas with necrosis and rarefaction but reactive gliosis is more common. Taken together, these lesions are suggestive of a longer (subacute) infection of the brain.

The following statements have been added: *“These microscopic findings are all more suggestive of a longer, subacute, infection of the brain”* and *“On necropsy, the dolphin had severe ulcerative inflammation of the pharynx with extensive loss of epithelium extending from the caudal tongue past the larynx. Histologically these areas demonstrated moderate to severe mucosal autolysis and no evidence of virus was found. Multiple levels of the gastrointestinal tract (forestomach, pyloric stomach, glandular stomach, proximal small intestine, middle small intestine, and colon) were also evaluated for the presence of influenza virus nucleoprotein by IHC. No definitive positive cells were detected in the intestinal mucosal. Gastrointestinal myenteric plexi were present and evaluated at all levels; they were all negative for virus antigen. More research needs to be done to determine how the virus spread to the CNS in this dolphin.”*

COMMENT #11. Aside for the relevance of this detection for humans and safety, I think the report is also important to consider cetacea might be at risk if the virus would manage to adapt.

RESPONSE: This is indeed a very relevant point and one we need to highlight. We have added the following paragraph to the discussion. *“Human health risk aside, the consequences of A(H5N1) viruses adapting for enhanced replication in and transmission between dolphins and other cetacea could be catastrophic for these populations. The molecular markers associated with influenza virus adaptation to these hosts is unknown, and, therefore, it is not possible to determine if there is any evidence of it occurring. Enhanced follow up of naturally occurring cetacea A(H5N1) infections with viral sequencing is prudent, with the possibility of identifying common mutations suggestive of host range markers.”*

COMMENT #12. Sequences have been submitted to GenBank?

RESPONSE: Yes, the sequences generated from the brain tissue were submitted to GenBank and the accession numbers OP698125-OP698132 are now available and added to the manuscript.

COMMENT #13. Methods line 313: please specify what authors consider ‘a complete set of tissues’.

RESPONSE: To provide further clarity here we have added the following description.

“A complete set of tissues were collected in 10% neutral buffered formalin: right cortex at caudate nucleus, right diencephalon at thalamus, right mesencephalon and hippocampus, right brainstem and cerebellum, spinal cord, liver, spleen, kidney, right and left adrenals, cranial omentum with debris, thyroid, right and left cranial lung, right and left middle lung, right caudal lung, right distal lung, trachea, tracheal lymph nodes, right and left tonsils, mandibular lymph nodes, cranial esophagus, laryngeal mucosa, laryngeal accessory lymph node, thymus, right and left atria, right and left ventricles, pulmonary artery, pulmonary marginal lymph node, interventricular septum, aorta, diaphragm, blubber and skin, urinary bladder, lumbar vertebral lymph node, glandular soft tissue, pancreas, tongue, pharynx, forestomach, pyloric stomach, glandular stomach, proximal intestine, pulmonary lymph hilar lymph node, middle intestine, mesenteric lymph node, distal intestine, colonic lymph node, prescapular lymph node, melon, testis, penis, right and left pterygoid sinus mucosa, and skeletal muscle and nerve.”

COMMENT #14. IHC and ISH: please provide information on the positive and negative controls used for both techniques. Polyclonal anti H1N1 apparently worked well. What was the reason for choosing this antibody?

RESPONSE: We have added the following details of the controls to the revised manuscript.

“Infected mouse lung was used as the positive control tissue and normal goat serum as the negative control primary antibody.” and *“The universal negative control probes developed by ACDBio, which target the DapB gene (accession # EF191515) from Bacillus subtilis strain SMY, a soil bacterium were used as controls in ISH.”*

We apologize for omitting the fact that the H1N1 antibody was generated to the H1N1 virus NP protein which is broadly conserved (determined by our many years of its use against many different subtypes of influenza virus). This has also now been added to clarify.

COMMENT #15. Can the authors say something about how sensitive this IHC is compared to a monoclonal anti influenza nucleoprotein e.g.?

RESPONSE: Unfortunately, we have not compared the sensitivity of this polyclonal antibody-based IHC assay with any monoclonal anti-influenza nucleoprotein antibodies directed against influenza nucleoprotein.

COMMENT #16. Fig 2: the bar length indications can’t be correct. Please check.

RESPONSE: We have changed C and D in Figure 2 to μm instead of mm.

Reviewer #2 (Remarks to the Author):

In the manuscript entitled “Highly pathogenic avian influenza A(H5N1) virus in a common bottlenose dolphin 2 (*Tursiops truncatus*) in Florida”, Dr Murawski and colleagues provide a detailed report about a common bottlenose dolphin in Florida that was infected with HPAI A(H5N1) clade 2.3.4.4b. Some basic information was already provided in the press release (A first: Avian influenza detected in American dolphin » College of Veterinary Medicine » University of Florida (ufl.edu)), but this report provides an interesting and thorough description of the pathological findings, PCR results, phylogenetic analysis, NAI susceptibility and receptor affinity. Furthermore, experiments are well performed, although the phylogenetic analysis is somewhat basic.

COMMENT #1. in the legend of figure 3, it is unclear why some virus names are indicated in green.

RESPONSE: We apologize for this oversight and overlooking the different colors in the phylogenetic tree. We have updated the caption to highlight that the HA sequences colored green are the most similar to that of the bottlenose dolphin, which is represented in blue. These sequences correspond to the green circles and blue square on the tree. Furthermore, the different colored rectangles highlight the different gene sources due to reassortment of viruses identified in North America.

We have updated the caption to say “**Fig. 3. Phylogenetic alignment of influenza A/bottlenose dolphin/Florida/UFTt2203/2022 (H5N1) HA gene sequences with Eurasian and North American lineage viruses.** Phylogenetic tree representing HA segments primarily from viruses collected from North American birds inferred using neighbor joining methods displayed with all 8 gene segment origins. The HA from the virus identified in the bottlenose dolphin is represented by the blue square while the green circles indicate the most similar sequences identified which differ by three nucleotides. Gene segments originating from North American viruses are represented in green, Eurasian origin gene segments are colored blue, and white indicates no data available.”

COMMENT #2. Furthermore, it might be of interest to provide as supplemental figures phylogenetic analysis of the different gene segments of A/bottlenose dolphin/Florida/UFTt2203/2022 and closely related viruses.

RESPONSE: We agree with the reviewer that the phylogenies of the other gene segments are interesting and have included the trees as supplemental figures. Due to the numerous reassortants identified in North America after the introduction of these A(H5N1) viruses, the phylogenies of each of the gene segments is crucial to understand the source of the viruses as they infect new hosts as well as the evolutionary potential within novel host species.

COMMENT #3. In addition, considering the very high viral loads in the brain, were any attempts performed to obtain the complete genome directly from the clinical sample?

RESPONSE: Thank you for the comment. We did indeed obtain full genome sequences from the brain homogenate as well as viral isolates from eggs and MDCK cells. We have updated the text as follows to clarify; “Sequence analysis of the full genome of A/bottlenose dolphin/Florida/UFTt2203/2022 (H5N1) virus was obtained directly from the brain homogenate as well as from virus isolated in embryonated chicken eggs and MDCK cells inoculated with homogenates from the dolphin brain. The genomic sequences from all sources were identical at the amino acid level.”

REVIEWERS' COMMENTS:

Reviewer #1 (Remarks to the Author):

Thank you for the revised manuscript and addressing all of my comments so extensively.

I have only some very minor suggestions:

I would move the IHC results of the intestines that are now in the discussion (line 280-285) to the results section discussing IHC (now of lung and brain only) 'Multiple levels of the gastrointestinal tract (forestomach, pyloric stomach, glandular stomach, proximal small intestine, middle small intestine, and colon) were also evaluated for the presence of influenza virus nucleoprotein by IHC. No definitive positive cells were detected in the intestinal mucosal. Gastrointestinal myenteric plexi were present and evaluated at all levels; they were all negative for virus antigen.' In the discussion you could suffice to shortly repeat those results (e.g. intestines were negative for virus antigen at all levels), so the port of entre of the infection remains unclear in this case.)

For the methods of the IHC two additional really minor points:

- 'infected mouse lung': as you use an H1N1 antibody, for detecting H5N1 nucleoprotein, to be concise and complete, I suggest you add the virus that the mouse was infected with. E.g. H1N1 mouse adapted virus infected mouse lung or whatever it was infected with.

- I don't understand how your goat serum is a negative control. I guess you might have used it to block antigens in the tissues prior to using your primary antibody. But that is not what I meant. A negative control can be an isotype control (only using the labelled secondary antibody, but not the primary, on the same tissue block), or it can be the (lung and brain) tissues of a dolphin that tested negative for influenza and letting that go through the same IHC protocol.

I trust the results, but I do think we need to be careful with IHC and as a standard always add your controls and would like to see those described in the methods.

Reviewer #2 (Remarks to the Author):

The authors have responded to all my comments very well.

RESPONSES TO REVIEWER'S COMMENTS ON MANUSCRIPT COMMSBIO-23-2140-T PART 2

Highly pathogenic avian influenza A(H5N1) virus in a common bottlenose dolphin (*Tursiops truncatus*) in Florida

Allison Murawski, Thomas Fabrizio, Robert Ossiboff, Christina Kackos, Trushar Jeevan, Jeremy C. Jones, Ahmed Kandeil, David Walker, Jasmine C. M. Turner, Christopher Patton, Elena A. Govorkova, Helena Hauck, Suzanna Mickey, Brittany Barbeau, Reddy Bommineni, Mia Torchetti, Lisa Kercher, Andrew B. Allison, Peter Vogel, Richard J. Webby, Michael Walsh

REVIEWER #1 (Remarks to the Author):

Thank you for the revised manuscript and addressing all of my comments so extensively. I have only some very minor suggestions: I would move the IHC results of the intestines that are now in the discussion (line 280-285) to the results section discussing IHC (now of lung and brain only) 'Multiple levels of the gastrointestinal tract (forestomach, pyloric stomach, glandular stomach, proximal small intestine, middle small intestine, and colon) were also evaluated for the presence of influenza virus nucleoprotein by IHC. No definitive positive cells were detected in the intestinal mucosal. Gastrointestinal myenteric plexi were present and evaluated at all levels; they were all negative for virus antigen.' In the discussion you could suffice to shortly repeat those results (e.g. intestines were negative for virus antigen at all levels), so the port of entre of the infection remains unclear in this case.)

For the methods of the IHC two additional really minor points:

- 'infected mouse lung': as you use an H1N1 antibody, for detecting H5N1 nucleoprotein, to be concise and complete, I suggest you add the virus that the mouse was infected with. E.g. H1N1 mouse adapted virus infected mouse lung or whatever it was infected with.

- I don't understand how your goat serum is a negative control. I guess you might have used it to block antigens in the tissues prior to using your primary antibody. But that is not what I meant. A negative control can be an isotype control (only using the labelled secondary antibody, but not the primary, on the same tissue block), or it can be the (lung and brain) tissues of a dolphin that tested negative for influenza and letting that go through the same IHC protocol.

I trust the results, but I do think we need to be careful with IHC and as a standard always add your controls and would like to see those described in the methods.

COMMENT #1: Please include a few sentences on the IHC results of the intestines in the results section.

RESPONSE #1:

The following sentences from discussion were moved to the results section line 159: "multiple levels of the gastrointestinal tract (forestomach, pyloric stomach, glandular stomach, proximal small intestine, middle small intestine, and colon) were also evaluated for the presence of influenza virus nucleoprotein by IHC. No definitive positive cells were detected in the intestinal mucosal. Gastrointestinal myenteric plexi were present and evaluated at all levels; they were all negative for virus antigen."

The above section was removed from discussion and replaced with “gastrointestinal sections were also negative for influenza virus nucleoprotein by IHC, so the port of entry for the virus remains unclear in this case.” (line 284-286).

Comment #2: Please clarify what virus strain/isolate the mouse was infected with in the methods of the IHC.

RESPONSE #2: The positive control mouse lungs used in this study were from 6-to-8-week-old female BALB/c H5N1 infected mice (Jackson Laboratory, Bar Harbor, ME, USA) (Kandeil, Nat. Comm., 2023, PMID: 37248261). The polyclonal goat antibody was shown to specifically label infected cells in mouse lungs.

This change was added to line 419.

COMMENT #3: Please also include the negative controls that the Reviewer 1 references (negative staining on uninfected dolphin tissue) by either describing it in more details in the methods, or as a supplemental figure.

RESPONSE #3:

We used the normal goat serum control to rule out any non-specific binding of goat antibodies to formalin-fixed dolphin tissues and did not see any evidence of non-specific binding with the normal goat serum. For the previous 20+ years, investigators at St Jude have used the goat polyclonal antibody described in this report to detect a wide range of influenza virus A subtypes in infected tissues from multiple species (primarily ferret and mouse). In our experience, and according to the supplier, this polyclonal antibody is H1N1 specific by IHA, but reacts with all antigenic types of Influenza A in other assay applications (including IHC). It does not react Influenza B, RSV, Para 1-3 or Adenovirus.

As described in our report, we observed multiple foci in the brain and meninges of the dolphin that were IHC positive. However, since we also saw extensive weaker labeling of white tracts that appeared to be non-specific, we decided that it was essential to confirm the IHC labeling with a virus-specific in situ hybridization assay. Using sequential sections, we used ISH to confirm that the IHC positive areas in the grey matter (neurons) and meninges were also H5N1 virus-specific by ISH. The ISH assay also confirmed that the weak IHC labeling of white tracts represented a non-specific immunohistochemical cross-reaction.

We have previously recognized that this goat polyclonal antibody raised against Influenza A Strain A/USSR/90/77 (H1N1) produced non-specific labeling of white tracts and optic nerves in ferret tissues, and now in dolphin as well. This has not been a problem with most other Influenza A subtypes, which are not neurotropic. However, given the increasing importance of neurovirulent H5N1 strains in the CNS, the results of this study and our studies in ferrets spurred us to identify a more specific primary antibody for IHC studies. We therefore identified and compared 7 different commercially available rabbit monoclonal antibodies that are directed against viral nucleoproteins of several different virus subtypes.

We tested these 7 different rabbit monoclonal antibodies against a wide range of virus subtypes in both lung and brain of infected mice and ferrets and identified two antibodies that labeled infected cells in the lung, nose, and brain, while completely avoiding non-specific labeling of white tracts in the brain. According to the supplier, the rabbit monoclonal antibody (clone name: HL1089) that we selected was raised against a recombinant protein comprising the central region of Influenza A virus Nucleoprotein (A/Kansas/14/2017(H3N2)). The supplier data sheet indicates that this monoclonal antibody is specific for a range of Influenza A virus Nucleoprotein proteins, including H1N1, H3N2, H5N8, and H10N3, and that it does not cross-react with Influenza B virus Nucleoprotein. Our results show that it also is specific for H5N1 virus Nucleoprotein, and we recently discontinued use of the goat polyclonal antibody for our flu IHC studies. We use a rabbit IgG isotype negative control for these IHC experiments.

A shortened response to this comment was added to the reporting summary document under the antibody section for the general public to view.